# High Carbon Dioxide Concentration Inhibits Pileus Growth of *Flammulina velutipes* by Downregulating Cyclin Gene Expression

**DOI:** 10.3390/jof11080551

**Published:** 2025-07-24

**Authors:** Kwan-Woo Lee, Che-Hwon Park, Seong-Chul Lee, Ju-Hyeon Shin, Young-Jin Park

**Affiliations:** 1Chungcheongbuk-do Agricultural Research and Extension Service, 46 Gagok-gil, Ochang-eup, Cheongwon-gu, Cheongju-si 28130, Republic of Korea; toolia18@korea.kr; 2Department of Medicinal Biosciences, Research Institute for Biomedical & Health Science, College of Biomedical and Health Science, Konkuk University, 268 Chungwon-daero, Chungju-si 27478, Republic of Korea; chehwon9798@kku.ac.kr (C.-H.P.); vwm963@kku.ac.kr (S.-C.L.); shin99@kku.ac.kr (J.-H.S.)

**Keywords:** carbon dioxide, cell cycle, cyclin, *Flammulina velutipes*, pileus

## Abstract

*Flammulina velutipes* is a widely cultivated edible mushroom in East Asia, recognized for its nutritional benefits and distinct morphology characterized by a long stipe and a compact, hemispherical pileus. The pileus not only plays a critical biological role in reproduction through spore formation but also serves as a key commercial trait influencing consumer preference and market value. Despite its economic importance, pileus development in *F. velutipes* is highly sensitive to environmental factors, among which carbon dioxide (CO_2_) concentration is particularly influential under indoor cultivation conditions. While previous studies have reported that elevated CO_2_ levels can inhibit pileus expansion in other mushroom species, the molecular mechanisms by which CO_2_ affects pileus growth in *F. velutipes* remain poorly understood. In this study, we investigated the impact of CO_2_ concentration on pileus morphology and gene expression in *F. velutipes* by cultivating fruiting bodies under two controlled atmospheric conditions: low (1000 ppm) and high (10,000 ppm) CO_2_. Morphometric analysis revealed that elevated CO_2_ levels significantly suppressed pileus expansion, reducing the average diameter by more than 50% compared to the low CO_2_ condition. To elucidate the underlying genetic response, we conducted RNA sequencing and identified 102 differentially expressed genes (DEGs), with 78 being downregulated under elevated CO_2_. Functional enrichment analysis highlighted the involvement of cyclin-dependent protein kinase regulatory pathways in this response. Two cyclin genes were found to be significantly downregulated under elevated CO_2_ conditions, and their suppression was validated through quantitative real-time PCR. These genes, possessing conserved cyclin_N domains, are implicated in the regulation of the eukaryotic cell cycle, particularly in mitotic growth. These results indicate that CO_2_-induced downregulation of cyclin genes may underlie cell cycle arrest, contributing to inhibited pileus development. This study is the first to provide transcriptomic evidence that elevated CO_2_ concentrations specifically repress PHO80-like cyclin genes in *F. velutipes*, revealing a molecular mechanism by which CO_2_ stress inhibits pileus development. These findings suggest that elevated CO_2_ triggers a morphogenetic checkpoint by repressing PHO80-like cyclins, thereby modulating cell cycle progression during fruiting body development. This study provides the first evidence of such a transcriptional response in edible mushrooms and offers promising molecular targets for breeding CO_2_-resilient strains and optimizing commercial cultivation conditions.

## 1. Introduction

*Flammulina velutipes*, commonly known as the winter mushroom or enokitake, is a commercially important edible basidiomycete belonging to the family *Physalacriaceae.* It is extensively cultivated in East Asian countries, particularly in Republic of Korea, Japan, and China, due to its high nutritional value, subtle flavor, and desirable texture. The species is characterized by a long, slender stipe and a small, dome-shaped pileus, which is a reproductive structure responsible for spore production and dispersal [1,2,3]. Among the morphological traits, the size and shape of the pileus are considered critical determinants of both reproductive fitness and commercial value [1,2,3].

The global demand for *F. velutipes* has grown steadily over the past few decades due to increasing consumer interest in functional foods and natural health products. Rich in dietary fiber, antioxidants, and immunomodulatory compounds such as polysaccharides and lectins, *F. velutipes* has garnered attention not only in gastronomy but also in nutraceutical and pharmaceutical industries [4,5]. Consequently, optimizing cultivation practices to improve yield and morphological uniformity has become a focal point in mushroom biotechnology.

Among the numerous environmental variables influencing mushroom cultivation—such as temperature, humidity, photoperiod, and substrate composition—gas composition, particularly the concentration of CO_2_, has emerged as a key factor that modulates the morphogenesis of fungal fruiting bodies [6,7,8,9,10,11]. In intensive cultivation systems, especially those employing sealed or semi-sealed containers, CO_2_ accumulation is common and can significantly affect the developmental trajectory of mushrooms. Elevated CO_2_ levels have been reported to suppress pileus expansion and alter the differentiation of various mushroom species, leading to reduced aesthetic quality and lower marketability [6,7,8,9,10,11].

Although the inhibitory effects of elevated CO_2_ on pileus development have been previously observed in species such as *Pleurotus ostreatus* and *F. filiformis*, the underlying molecular mechanisms remain incompletely understood [7,11]. In these species, transcriptomic and proteomic studies suggest that elevated CO_2_ may interfere with genes and proteins involved in the cell cycle, cell wall biosynthesis, and energy metabolism [7,11]. However, despite the agricultural and economic importance of *F. velutipes*, detailed investigations into its transcriptional response to CO_2_ stress have been limited. No studies to date have conclusively identified the key regulatory genes mediating pileus suppression under elevated CO_2_ conditions in *F. velutipes*.

Given the growing need for precision agriculture in fungal biotechnology, there is a pressing demand to understand how environmental cues like CO_2_ concentration affect gene expression and developmental pathways in commercially valuable mushrooms. Such knowledge could facilitate the development of CO_2_-resilient strains through targeted breeding or genetic engineering, thereby enhancing productivity and quality under suboptimal cultivation conditions.

In this study, we investigated the effects of CO_2_ concentration on the morphological development of *F. velutipes*, with a particular focus on pileus size. We cultivated *F. velutipes* under two controlled CO_2_ conditions—1000 ppm (ambient) and 10,000 ppm (elevated)—and performed a comparative transcriptomic analysis to identify differentially expressed genes (DEGs). Functional enrichment analyses revealed that two cyclin genes, known for their pivotal role in cell cycle regulation, were significantly downregulated under elevated CO_2_ conditions. Quantitative PCR confirmed the downregulation of cyclin genes under elevated CO_2_, consistent with RNA-Seq results. Accordingly, this study was designed to explore the transcriptomic mechanisms of CO_2_-induced morphological changes in *F. velutipes*, aiming to identify key genes involved in developmental regulation and provide a foundation for future strain improvement efforts.

## 2. Materials and Methods

### 2.1. Fungal Strain and Cultivation Conditions

The *F. velutipes* strain CBMFV-72 was obtained from the Agricultural Research and Extension Service, Cheongju-si, Republic of Korea. This strain was selected based on its stable morphological traits and widespread use in commercial cultivation studies. The mycelia were initially cultured on potato dextrose agar (PDA; Difco, Seoul, Republic of Korea) at 25 °C for 20 days to promote vegetative growth. Following the pre-culture stage, actively growing mycelial plugs were transferred to a solid fruiting substrate composed of 80% sawdust and 20% rice bran, which was sterilized and incubated under controlled environmental conditions. The incubation for mycelial colonization was carried out at 18 °C with 62–65% relative humidity for 25 days in darkness. Subsequently, to induce fruiting body development, the temperature was sequentially reduced to 14–16 °C under 80% humidity and exposed to light (approximately 300 lux) for 8 h per day.

Two different CO_2_ concentrations were maintained during the fruiting phase using an automated gas regulation system incorporating a non-dispersive infrared (NDIR) sensor (SH-VT260; SOHA Tech, Seoul, Republic of Korea): a low CO_2_ condition (1000 ppm, simulating ambient atmospheric levels) and an elevated CO_2_ condition (10,000 ppm), which reflects accumulation levels frequently encountered in enclosed cultivation facilities. Gas levels were continuously monitored using infrared CO_2_ sensors to ensure stability. Fruiting bodies were collected at the stage of morphological maturity, characterized by full elongation of the stipe and initial pileus expansion.

### 2.2. Morphological Analysis

To assess the impact of CO_2_ concentration on pileus development, pileus diameter was measured from at least 30 fruiting bodies per treatment group using digital calipers. Data were statistically analyzed using Student’s *t*-test, with a significance threshold of *p* < 0.001.

### 2.3. RNA Extraction and Sequencing

Total RNA was extracted from freshly harvested pileus tissues, which were immediately cryopreserved in liquid nitrogen and ground to a fine powder using a pre-cooled mortar and pestle. RNA was isolated using TRIzol reagent (Thermo Fisher Scientific, Seoul, Republic of Korea) following the manufacturer’s instructions, and further purified using the RNeasy Mini Kit (Qiagen, Seoul, Republic of Korea) with on-column DNase I treatment to remove genomic DNA contamination. RNA quality and concentration were assessed using a NanoDrop spectrophotometer (Thermo Fisher Scientific, Seoul, Republic of Korea) and agarose gel electrophoresis. High-quality RNA samples (RIN > 7.0) were selected for sequencing.

We constructed RNA-Seq libraries from 1 μg of total RNA and sequenced them using the Illumina HiSeq 2000 platform (Illumina Korea, Seoul, Republic of Korea), generating paired-end reads. Raw reads were trimmed for quality using a Phred score cutoff of 30 and processed using the Trinity de novo assembly pipeline (v2.15.0) [12]. TransDecoder (v5.5.0) [13] was used to predict coding regions, and CD-HIT (v4.8.1) [14] was employed to reduce redundancy through clustering and deduplication of transcripts (Appendix A).

### 2.4. Differential Gene Expression and Enrichment Analysis

Differentially expressed genes (DEGs) between the low and elevated CO_2_ groups were identified using DESeq2 (v1.48.1) integrated within the Trinity pipeline [15]. The thresholds for significance were set at |log_2_ fold change| > 1 and *p* < 0.001. Functional annotation and enrichment analysis of DEGs were performed using the STRING database (v12.0) [16] and the topGO R package (v2.54.0) [17]. GO terms were assigned across three categories: biological process, molecular function, and cellular component. Genes were annotated with domain information using Pfam (24 February 2024) [18] and NCBI Conserved Domain Database (CDD) searches [19].

### 2.5. Quantitative Real-Time PCR Validation

To validate the RNA-Seq results, quantitative real-time PCR (qRT-PCR) was performed on a subset of differentially expressed genes identified under elevated CO_2_ conditions. Total RNA was extracted separately from (i) whole fruiting bodies and (ii) dissected pileus and stipe tissues to assess possible tissue-specific expression patterns. In mushrooms, the pileus is the cap-like upper structure that expands and bears the spore-producing hymenium, whereas the stipe is the stem-like supporting structure that elevates the pileus. First-strand cDNA synthesis was carried out using 1 μg of total RNA, oligo(dT) primers, 5× RT buffer, and M-MLV Reverse Transcriptase (Bioneer, Daejeon, Republic of Korea). The reaction mixture was incubated at 25 °C for 5 min, followed by 42 °C for 60 min.

qPCR was conducted using the Bioline SensiFAST SYBR No-ROX kit (Joagene Bioscience, Seoul, Republic of Korea) on a Rotor-Gene 6000 instrument (Qiagen). Each reaction contained 100 ng cDNA, 10 pM gene-specific primers, and SYBR master mix in a final volume of 20 µL. The primer sequences used for amplification are listed in Appendix A. Gene expression levels were normalized to an internal reference gene, and relative expression was calculated using the ^ΔΔCt^ method. qRT-PCR was performed with three biological replicates, each with three technical replicates. Statistical analysis was performed using *t*-tests to evaluate significant differences between CO_2_ treatments.

## 3. Results

### 3.1. Elevated CO_2_ Concentration Significantly Reduces Pileus Size in F. velutipes

The morphological assessment of *F. velutipes* cultivated under two CO_2_ concentrations—1000 ppm (low) and 10,000 ppm (high)—revealed a pronounced effect of elevated CO_2_ on pileus development. As shown in Figure 1a, fruiting bodies grown under elevated CO_2_ conditions displayed a visibly smaller and less expanded pileus compared to those grown under low CO_2_. However, the stipe length did not show a statistically significant difference between CO_2_ treatments. Quantitative measurements confirmed that the mean pileus diameter under elevated CO_2_ was 4.82 ± 1.11 mm, significantly smaller than the 10.23 ± 2.55 mm observed under low CO_2_ conditions (*p* < 0.001, *t*-test) (Figure 1b). These results are consistent with previous studies in *F. filiformis* and *Pleurotus ostreatus*, which reported inhibition of pileus expansion under elevated CO_2_ environments due to altered cellular proliferation and expansion mechanisms [7,11].

In *F. velutipes*, this reduction in pileus size further suggests that elevated CO_2_ may suppress normal fruiting body morphogenesis through changes in physiological or transcriptional regulation. Considering the pileus is the site of spore production and a key trait for commercial evaluation, this CO_2_-induced morphological alteration presents both biological and industrial concerns for mushroom growers [6,7,11].

### 3.2. CO_2_-Induced Transcriptomic Changes in F. velutipes

To explore the molecular mechanisms associated with CO_2_-induced morphological changes, RNA-Seq analysis was performed, resulting in the detection of 17,038 transcripts across all samples. A comparative transcriptomic analysis was then conducted to identify CO_2_-responsive gene expression patterns and assess their potential functional relevance (Appendix A). Differential gene expression analysis using DESeq2 identified significant transcriptomic alterations, with 20 top-ranked differentially expressed genes (DEGs) selected based on *p*-value < 0.01 and |log_2_ fold change| ≥ 1 (Figure 2, Table 1 and Appendix A). Among the upregulated genes under elevated CO_2_, transcripts such as DN3185_c0_g1_i20, DN95_c0_g1_i7, and DN985_c0_g1_i3 exhibited strong induction, with log_2_ fold changes exceeding seven. These genes are predicted to encode proteins involved in methyltransferase activity, heat shock protein function, and redox metabolism, respectively, suggesting enhanced cellular remodeling and stress adaptation [20,21,22]. Conversely, transcripts including DN1803_c0_g1_i17 and DN4990_c0_g1_i119 were among the most downregulated genes under elevated CO_2_. These transcripts are associated with major facilitator superfamily transporters and SPX (SYG1, PHO81, and XPR1) domain-containing proteins, potentially linked to nutrient signaling or membrane transport, both of which may be suppressed in carbon-rich environments [23,24]. Notably, genes predicted to encode enzymes such as cytochrome P450s (e.g., DN28_c0_g1_i39, DN5300_c0_g1_i5) were differentially expressed, implicating a role for oxidative metabolism in CO_2_ response. Several DEGs belonged to protein families involved in structural regulation, such as ankyrin repeats and WD40 domains, which are often associated with cytoskeletal organization and developmental signaling [25,26]. These transcriptional changes mirror phenotypic responses observed in *F. velutipes* under elevated CO_2_, including enhanced stipe elongation and delayed cap development. Such morphological alterations are commonly leveraged in commercial cultivation to optimize fruiting body appearance [6]. The upregulation of genes potentially involved in cell wall modification (e.g., chitin biosynthesis, β-glucan remodeling) and stress response indicates a shift toward promoting vegetative growth and suppressing reproductive development [27,28,29]. Furthermore, downregulated genes were enriched in pathways potentially related to aerobic respiration, consistent with metabolic adjustments under limited oxygen availability and excess CO_2_ [22,30,31,32]. Together, these findings suggest that *F. velutipes* reprograms its gene expression landscape in response to CO_2_ levels, modulating energy metabolism, stress tolerance, and developmental timing.

### 3.3. CO_2_-Responsive Transcriptomic Profiling of CAZyme Genes

Genome-wide analysis of *F. velutipes* revealed a diverse array of carbohydrate-active enzymes (CAZymes), including glycoside hydrolases (GHs), auxiliary activities (AAs), carbohydrate esterases (CEs), polysaccharide lyases (PLs), and carbohydrate-binding modules (CBMs), underscoring its capacity for efficient lignocellulosic biomass degradation [1]. CAZyme-related genes were predicted based on annotation using the CAZy database (14 March 2024) (Appendix A) [33]. Prominent GH families such as GH5, GH16, and GH43 were identified, which are known to participate in the breakdown of cellulose and hemicellulose, while multiple AA9 genes encoding lytic polysaccharide monooxygenases (LPMOs) suggest a strong oxidative degradation capability that synergizes with hydrolytic enzymes (Appendix A) [34]. The presence of CE and PL genes further implies the ability to remove complex side chains and target diverse polysaccharide components, whereas CBM-containing enzymes—particularly those with CBM1 and CBM13—likely enhance substrate binding and catalytic efficiency [34]. Signal peptide predictions in many CAZyme genes suggest their secretion into the extracellular environment, aligning with the saprotrophic lifestyle of *F. velutipes* [34].

In addition to genomic potential, transcriptomic profiling under varying CO_2_ concentrations revealed dynamic expression changes in response to environmental carbon levels. Notably, six CAZyme-related genes exhibited significant differential expression between high and low CO_2_ conditions, indicating a transcriptional regulatory mechanism responsive to atmospheric cues (Figure 3a and Table 2) [35]. These included genes encoding GH114 and GH71 enzymes, which are involved in the degradation of α-glucans and β-glucans, respectively, as well as AA9-type LPMOs that mediate oxidative cleavage of cellulose [33]. Two additional genes encoding CE5 and PL38 family proteins also showed significant expression changes, suggesting CO_2_ sensitivity not only in catalytic enzymes but also in polysaccharide lyases. Notably, a gene containing both CBM49 and GH13_S domains was also differentially expressed, indicating possible involvement in starch-related substrate recognition. Interestingly, the highly upregulated genes under low CO_2_ concentrations included those related to hemicellulose and pectin degradation, implying that *F. velutipes* may prioritize the breakdown of accessible carbohydrates under limited CO_2_ conditions as a strategy to balance energy demand and metabolic flexibility [35]. This CO_2_-responsive expression pattern highlights the plasticity of the fungal degradative system and suggests that environmental CO_2_ not only acts as a substrate-related cue but may also serve as a signaling molecule influencing carbon acquisition strategies. While definitive evidence is still lacking, the observed upregulation of CAZyme genes under low CO_2_ conditions may be indicative of enhanced structural plasticity and potential expansibility of the fruiting body. This expression pattern could reflect an adaptive developmental strategy whereby *F. velutipes* modulates its cell wall remodeling capacity in response to environmental signals, thereby optimizing morphogenesis under fluctuating atmospheric conditions.

To validate these transcriptomic findings, we performed qRT-PCR for six representative CAZyme genes, confirming significant expression differences between 1000 ppm and 10,000 ppm CO_2_ conditions (Figure 3b). These results support the conclusion that CAZyme expression is transcriptionally regulated by atmospheric CO_2_ levels. These findings have implications for fungal adaptation in changing atmospheric environments and could be leveraged for optimizing biomass degradation in industrial applications under controlled CO_2_ conditions.

### 3.4. Transcriptome Profiling Reveals Downregulation of Cell Cycle-Related Genes Under Elevated CO_2_ Conditions

Out of the 17,038 transcripts detected by RNA-Seq, a total of 102 genes were identified as differentially expressed between high and low CO_2_ conditions, based on a *p*-value < 0.01 and |log_2_ fold change| ≥ 1 (Appendix A). Among these, 78 genes were downregulated and 24 were upregulated under elevated CO_2_ conditions. Gene Ontology (GO) enrichment analysis of the downregulated genes identified significant terms related to cell cycle regulation, particularly cyclin-dependent kinase (CDK) activity and protein kinase binding (Figure 4). Notably, two genes (DN1560_c0_g1_i9 and DN6954_c0_g1_i2) were commonly detected as members of the cyclin family in the enrichment analysis based on the translated protein sequences (Table 3). Pfam domain analysis further confirmed the presence of conserved cyclin_N box domains at the N-termini of these proteins (residues 62–157). In addition, comparative sequence analysis using the NCBI Conserved Domain Database (CDD) analysis revealed that both proteins share high sequence similarity with members of the cyclin_SF superfamily in *Saccharomyces cerevisiae*, particularly with G1/S-specific cyclins such as PCL1 and PCL2, which are known to interact with the Pho85 cyclin-dependent kinase and regulate the G1/S phase transition in the eukaryotic cell cycle (Appendix A and Appendix A) [36,37]. In contrast, GO enrichment analysis of the upregulated genes did not yield any significantly enriched terms, suggesting that these genes may not be involved in coherent functional pathways or may participate in more diverse or poorly characterized processes.

The downregulation of cyclin genes under elevated CO_2_ conditions strongly suggests that elevated CO_2_ disrupts normal mitotic processes required for cell proliferation during pileus development. Cyclins are essential components of the cell cycle machinery, forming complexes with CDKs to regulate progression through various checkpoints [38,39]. In fungi, especially in basidiomycetes like *F. velutipes*, cell division and tissue differentiation are tightly coordinated during the formation of reproductive structures [40,41,42]. Suppression of these regulatory genes could impair cellular growth in the pileus region, leading to reduced expansion and final size.

### 3.5. Evolutionary Conservation of Cyclin Sequences Among Basidiomycota

To support the transcriptomic findings indicating CO_2_-responsive suppression of cyclin genes, we conducted a comprehensive investigation into the evolutionary conservation and domain architecture of the identified cyclin candidates using PSI-BLAST (26 March 2024) (threshold 0.005) and InterPro protein family classification (*e*-value < 0.001). Homology searches against fungal protein databases (NCBI non-redundant database) revealed that the majority of top cyclin homologs were derived from members of the phylum Basidiomycota, including the genera *Lentinula*, *Armillaria*, *Mycena*, and *Pleurotus* (Appendix A), suggesting that cyclin-mediated cell cycle regulation is evolutionarily conserved among mushroom-forming fungi.

Multiple sequence alignment of these homologs revealed strong conservation within the canonical cyclin box, which is thought to mediate interactions with cyclin-dependent kinases (CDKs) [43]. Although detailed structural data are limited for fungal cyclins, residues such as leucine (L), glutamic acid (E), and phenylalanine (F) are commonly involved in hydrophobic interactions in cyclin–CDK complexes in other eukaryotes (Figure 5) [44]. Their conservation across Basidiomycota homologs suggests a potential functional role in CDK association and maintenance of cyclin structure.

InterPro classification of the *F. velutipes* transcriptome identified seven transcripts belonging to the PHO80-like cyclin protein family (IPR013922), which includes a group of eukaryotic cyclins involved in regulating the G1/S phase transition of the cell cycle (Table 4) [45,46,47]. Among these, only two transcripts (DN1560_c0_g1_i9 and DN6954_c0_g1_i2) were significantly downregulated under elevated CO_2_ conditions and were concurrently enriched in Gene Ontology (GO) terms related to “regulation of CDK activity” and “protein kinase binding”. These two transcripts were the only PHO80-like cyclins to satisfy both differential expression and functional enrichment thresholds, indicating their unique regulatory roles in the CO_2_-responsive gene network. Moreover, comparative sequence analysis among PHO80-like cyclin genes within the *F. velutipes* genome revealed that only these two genes shared specific conserved sequence regions (outside the canonical cyclin_N domain), which are also commonly found across other Basidiomycota species (Appendix A), highlighting their evolutionary conservation and suggesting they act as key regulators of cell cycle control and environmental adaptation in response to CO_2_. Notably, DN1560_c0_g1_i9 was identified as an isoform from a single gene locus, and only this isoform was differentially expressed under elevated CO_2_. This suggests that structural differences between isoforms may affect their transcriptional responsiveness, highlighting the regulatory importance of transcript diversity in environmental adaptation.

The exclusive identification of DN1560_c0_g1_i9 and DN6954_c0_g1_i2 as core members of the PHO80-like family with environmental responsiveness suggests a high degree of functional specialization. This specificity may reflect regulatory divergence at the transcriptional level, differential promoter sensitivity to environmental stimuli, or spatial–temporal expression patterns within developing tissues. While the PHO80-like family encompasses multiple cyclins with structural similarity, only these two genes appear to be functionally integrated into the CO_2_-regulated developmental framework of *F. velutipes*.

Taken together, the convergence of phylogenetic conservation, family-level classification, and transcriptional responsiveness under CO_2_ stress highlights DN1560_c0_g1_i9 and DN6954_c0_g1_i2 as pivotal regulators of cell cycle progression. These genes represent promising molecular targets for developing CO_2_-tolerant mushroom strains and provide valuable insights into how environmental factors modulate morphogenetic pathways in basidiomycetous fungi.

### 3.6. Validation and Tissue-Specific Regulation of CO_2_-Induced Cyclin Gene Repression

To validate the transcriptomic observation of cyclin gene repression under elevated CO_2_ (10,000 ppm), we conducted qRT-PCR targeting two PHO80-like cyclin genes, DN6954_c0_g1_i2 and DN1560_c0_g1_i9, using RNA isolated from three distinct tissues of *F. velutipes*: the pileus, stipe, and whole fruiting body.

As shown in Figure 6, DN1560_c0_g1_i9 was significantly downregulated across all tissue types, with reductions of approximately 72% in the pileus, 64% in the stipe, and 68% in the whole fruiting body under high CO_2_ conditions (*p* < 0.01). These findings suggest a broad, CO_2_-responsive role of this gene in cell cycle regulation during fruiting body development.

In contrast, DN6954_c0_g1_i2 exhibited tissue-specific repression. While its expression was significantly decreased in the stipe (~81%) and whole fruiting body (~88%) (*p* < 0.001), no significant change was observed in the pileus (*p* > 0.05). This suggests that DN6954_c0_g1_i2 may play a more specialized role in regulating mitotic activity in the stipe rather than in the pileus.

Interestingly, despite the lack of differential expression in the pileus, DN6954_c0_g1_i2 still exhibited strong downregulation at the whole-fruiting-body level. This discrepancy likely reflects the disproportionately large biomass contribution of the stipe relative to the pileus in *F. velutipes* [1]. Given that the stipe is elongated and dominates the total volume of the fruiting body, expression changes in this tissue can strongly influence overall transcript levels. Moreover, the stipe undergoes vigorous cell division and elongation during development, further supporting the idea that DN6954_c0_g1_i2 functions primarily in stipe-specific cell cycle control [6].

These tissue-dependent patterns of gene expression repression imply that *F. velutipes* differentially regulates cyclin genes in response to CO_2_ stress depending on anatomical location. While DN1560_c0_g1_i9 appears to be a global regulator affecting multiple tissues, DN6954_c0_g1_i2 may serve as a stipe-specific CO_2_-responsive gene involved in controlling longitudinal growth. This is consistent with the observed CO_2_-induced morphological changes, in which stipe elongation is promoted while pileus expansion is inhibited under elevated CO_2_ conditions [7].

Overall, our qRT-PCR validation confirms the RNA-Seq results and reveals a spatially resolved regulatory mechanism, where CO_2_ influences the transcriptional repression of key cyclin genes in a tissue-dependent manner. These findings suggest that DN6954_c0_g1_i2 and DN1560_c0_g1_i9 represent valuable molecular markers for breeding CO_2_-resilient mushroom strains, and they offer important insights into how atmospheric conditions modulate organ-specific growth processes in basidiomycetes. However, further functional characterization will be necessary to validate the tissue-specific roles of these genes and to elucidate the underlying regulatory mechanisms in response to CO_2_ stress.

## 4. Discussion

In general, high concentrations of CO_2_ inhibit the growth of many aerobic microorganisms, but wood-decaying basidiomycetes such as *F. velutipes* are known as capnophilic or facultative anaerobes that thrive in low-oxygen, high-carbon environments [22,48]. However, our results suggest that CO_2_ may have inhibitory effects when it exceeds a certain threshold, especially during sensitive developmental stages such as fruiting body formation [6,36].

Transcriptome analysis showed that high concentrations of CO_2_ downregulated the expression of key regulators of cell cycle progression, particularly PHO80-like cyclins, suggesting that they act as regulators of pileus size under abnormal atmospheric conditions. This may be an adaptive strategy to conserve energy in environments with limited ventilation such as decaying logs [9,22,49,50,51,52].

Interestingly, under high CO_2_ conditions, the growth of the pileus was suppressed, but the number of fruiting bodies significantly increased (Appendix A). Analysis of the number of fruiting bodies showed that *F. velutipes* cultured under 10,000 ppm CO_2_ produced significantly more fruiting bodies than those under 1000 ppm CO_2_, suggesting a shift to a compensatory developmental strategy that prioritizes increased reproductive output over pileus expansion. This morphological change may be a fruiting body development strategy for efficient spore dispersal and environmental adaptation in an oxygen-limited environment.

These results suggest that CO_2_ may act as a signal that affects not only pileus size but also the number of fruiting bodies in *F. velutipes*. Therefore, the suppression of cyclin expression and the associated morphological changes in *F. velutipes* may reflect an adaptive strategy to gas changes in the atmosphere. Although this study did not specifically clarify the ecological implications of environmental changes in *F. velutipes*, it provides an interesting hypothesis that should be examined in future studies.

In terms of application, this study also provides important implications for mushroom cultivation. In particular, in modern bottle or bag cultivation systems, CO_2_ accumulation due to insufficient ventilation causes pileus size variation and inconsistent product quality [6,9,53]. The CO_2_-responsive cyclin gene of *F. velutipes* identified in this study can be utilized as a candidate molecular marker for breeding high CO_2_-tolerant strains. In addition, it will be possible to improve cultivation conditions that maintain both yield and morphological quality through real-time environmental monitoring and air flow optimization.

In this study, transcriptome analysis of *F. veutipes* revealed a significant association between cyclin genes and changes in pileus size and number of fruiting bodies. However, further studies involving mutation or overexpression are needed to clarify the functional roles of these cyclin genes, and in parallel, future research should explore how elevated CO_2_ levels influence quality-related traits such as taste, texture, and nutritional composition, in addition to morphological changes.

## 5. Conclusions

In this study, we demonstrated that elevated atmospheric CO_2_ concentration (10,000 ppm) significantly inhibits pileus development in *F. velutipes* by downregulating two PHO80-like cyclin genes, DN1560_c0_g1_i9 and DN6954_c0_g1_i2, which are implicated in mitotic regulation during fruiting body formation. This repression was validated through RNA-Seq and qRT-PCR across multiple tissue types and shows strong homology to the *PCL1/PCL2* family of *S. cerevisiae*, suggesting conserved regulatory mechanisms across fungi.

Our results represent the first clear indication that repression of specific cyclin genes contributes to reduced pileus expansion under elevated CO_2_ in a cultivated mushroom species. These observations imply that CO_2_ may play a dual role—not only limiting development physically but also acting as a molecular signal that activates checkpoint pathways. Altogether, our results shed light on how fungi adjust their developmental processes at the transcriptomic level, and they may offer practical clues for enhancing traits in cultivated strains.

Furthermore, increased CO_2_ concentration led to a reduction in pileus size, accompanied by a compensatory rise in fruiting body number of *F. velutips*. This finding raises the possibility of a trade-off between pileus expansion and fruiting body proliferation under changing environmental conditions. This observation indicates the complexity of interpreting growth changes caused by the environment. It also emphasizes the need for an integrated understanding through physiological and molecular analyses in future studies.

Overall, these findings help us understand how environmental changes affect the development of mushroom-forming fungi and provide a foundation for developing CO_2_-tolerant mushroom cultivation strategies based on transcriptome-level insights.

## Figures and Tables

**Figure 1 jof-11-00551-f001:**
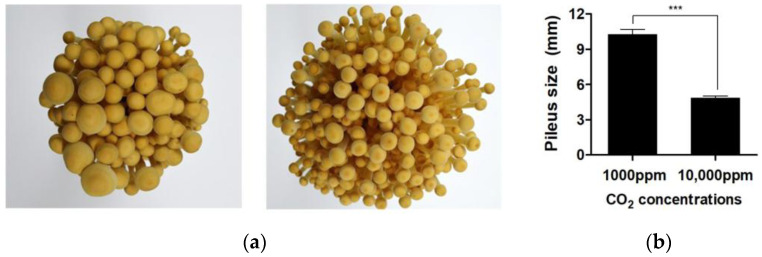
Effect of CO_2_ concentration on the pileus of *F. velutipes*. (**a**) Pileus of the fruiting bodies of *F. velutipes* cultivated under CO_2_ concentrations of 1000 ppm (**left**) and 10,000 ppm (**right**). (**b**) Pileus sizes of *F. velutipes* under different CO_2_ concentrations. Data were analyzed using *t*-tests. *** *p* < 0.001.

**Figure 2 jof-11-00551-f002:**
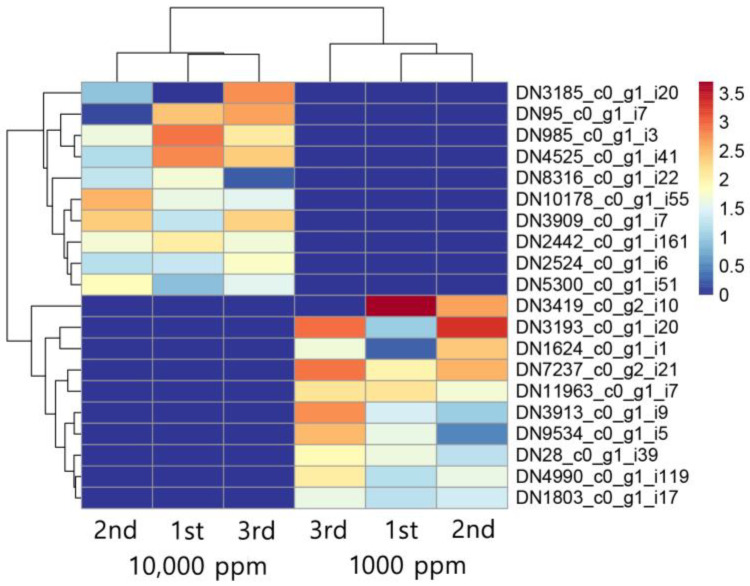
Top 10 upregulated and top 10 downregulated transcripts in *F. velutipes* in response to CO_2_ treatment (1000 ppm and 10,000 ppm). Heatmap shows hierarchical clustering of normalized expression levels across three biological replicates per condition.

**Figure 3 jof-11-00551-f003:**
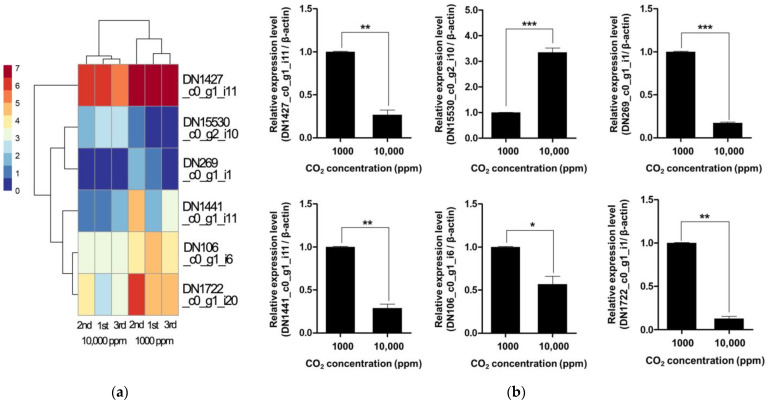
Heatmap (**a**) and qRT-PCR validation (**b**) of differentially expressed CAZyme genes in *F. velutipes* under low (1000 ppm) and high (10,000 ppm) CO_2_. Asterisks indicate statistically significant differences between treatments (* *p* < 0.05, ** *p* < 0.01, *** *p* < 0.001; Student’s *t*-test).

**Figure 4 jof-11-00551-f004:**
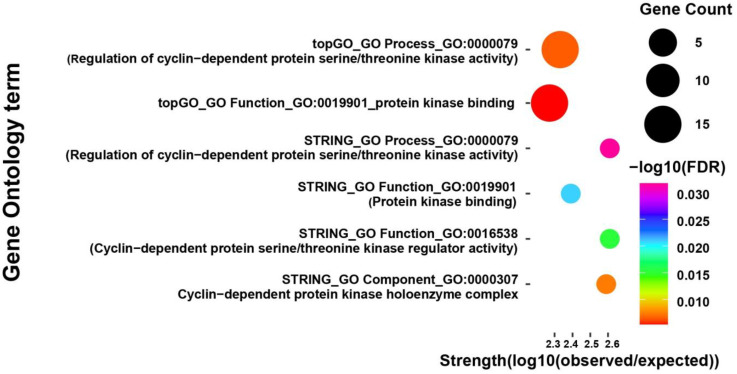
Gene Ontology (GO) enrichment analysis of downregulated genes in *F. velutipes* under elevated CO_2_ conditions (10,000 ppm). GO terms were identified using STRING and topGO analysis tools and are grouped into three categories: biological process, molecular function, and cellular component. Each bubble represents an enriched GO term. The size of the bubble corresponds to the number of genes associated with that term, the color scale indicates statistical significance as −log_10_ of the false discovery rate (FDR), and the *x*-axis represents enrichment strength.

**Figure 5 jof-11-00551-f005:**
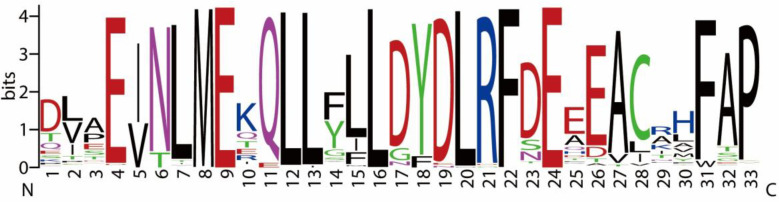
Conserved amino acid regions identified from PSI-BLAST alignment of *F. velutipes* PHO80-like cyclin homologs (DN1560_c0_g1_i9 and DN6954_c0_g1_i2) with cyclin sequences from other fungal species.

**Figure 6 jof-11-00551-f006:**
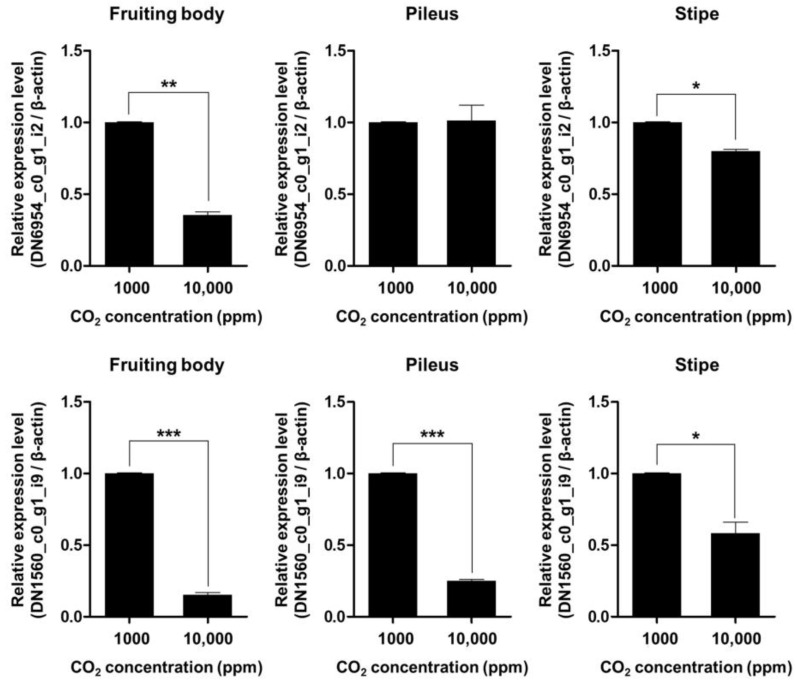
Quantitative real-time PCR validation of the relative expression levels of two PHO80-like cyclin genes (DN6954_c0_g1_i2 and DN1560_c0_g1_i9) in *F. velutipes* under different CO_2_ concentrations (1000 ppm and 10,000 ppm). Expression values were measured in the whole fruiting body, pileus, and stipe tissues. Bars represent mean ± standard error (*n* = 3 biological replicates). Asterisks indicate statistically significant differences between treatments (* *p* < 0.05, ** *p* < 0.01, *** *p* < 0.001; Student’s *t*-test).

**Table 1 jof-11-00551-t001:** Differentially expressed genes in *F. velutipes* under CO_2_ treatment (1000 ppm and 10,000 ppm). The table lists the top 10 downregulated and top 10 upregulated genes based on log_2_ fold change values. Gene expression levels are shown under two CO_2_ concentrations, and corresponding protein domain annotations are retrieved from InterPro (24 February 2024) and Pfam (24 February 2024) databases. An en dash (“–”) indicates no significant domain hit.

	Predicted Gene ID	FPKM ^1^		log_2_ Fold Change	InterPro Database		Pfam Database	
		10,000 ppm	1000 ppm					
					ID	Description	ID	Description
**Downregulated genes**	DN3193_c0_g1_i20	0.00	130.17	−9.50	IPR002554	Protein phosphatase 2A, regulatory B subunit, B56	PF01603.24	Protein phosphatase 2A regulatory B subunit (B56 family)
	DN3913_c0_g1_i9	0.00	81.14	−8.82	IPR045338	Domain of unknown function DUF6535	PF20153.3	Family of unknown function (DUF6535)
	DN3419_c0_g2_i10	0.00	49.18	−8.09	–	–	–	–
	DN7237_c0_g2_i21	0.00	36.92	−7.68	–	–	–	–
	DN28_c0_g1_i39	0.00	35.68	−7.63	IPR036396	Cytochrome P450 superfamily	PF00067.26	Cytochrome P450
	DN1624_c0_g1_i1	0.00	32.71	−7.51	–	–	PF01483.24	Proprotein convertase P-domain
	DN4990_c0_g1_i119	0.00	32.65	−7.51	–	–	PF03105.23	SPX domain
	DN1803_c0_g1_i17	0.00	28.99	−7.33	–	–	PF07690.20	Major Facilitator Superfamily
	DN11963_c0_g1_i7	0.00	27.47	−7.25	–	–	–	–
	DN9534_c0_g1_i5	0.00	26.86	−7.22	–	–	–	–
**Upregulated genes**	DN3909_c0_g1_i7	14.33	0.00	6.28	–	–	–	–
	DN5300_c0_g1_i51	19.79	0.00	6.75	–	–	PF00067.26	Cytochrome P450
	DN8316_c0_g1_i22	22.59	0.00	6.94	IPR015943	WD40/YVTN repeat-like-containing domain superfamily	PF10282.13	Lactonase, 7-bladed beta-propeller
	DN2442_c0_g1_i161	25.08	0.00	7.09	–	–	PF20152.3	Family of unknown function (DUF6534)
	DN4525_c0_g1_i41	27.28	0.00	7.21	IPR001663	Aromatic-ring-hydroxylating dioxygenase, alpha subunit	PF00355.30	Rieske [2Fe-2S] domain
	DN2524_c0_g1_i6	30.20	0.00	7.36	IPR036770	Ankyrin repeat-containing domain superfamily	PF13606.10	Ankyrin repeat
	DN985_c0_g1_i3	33.34	0.00	7.50	–	–	PF00724.24	NADH:flavin oxidoreductase/NADH oxidase family
	DN95_c0_g1_i7	39.11	0.00	7.73	–	–	PF00012.24	Hsp70 protein
	DN3185_c0_g1_i20	52.63	0.00	8.16	–	–	PF10017.13	Histidine-specific methyltransferase, SAM-dependent
	DN10178_c0_g1_i55	67.72	0.00	8.52	–	–	PF13639.10	Ring finger domain

^1^ Fragments Per Kilobase of transcript per Million mapped reads.

**Table 2 jof-11-00551-t002:** CAZyme-related genes in *F. velutipes* differentially expressed under CO_2_ treatment, selected based on log_2_ fold change thresholds (≥1 or ≤−1).

Predicted Gene ID	CAZyme	DB	Signalp ^1^	FPKM ^2^		log_2_ Fold Change
				10,000 ppm	1000 ppm	
DN1427_c0_g1_i11	AA9	DIAMOND	N	1865.41	3876.84	−1.06
DN15530_c0_g2_i10	GH114	HMMER, dbCAN_sub	Y(1–22)	38.65	6.22	2.63
DN269_c0_g1_i1	CE8	DIAMOND	N	0.00	26.33	−7.19
DN1441_c0_g1_i11	GH71	HMMER, dbCAN_sub, DIAMOND	N	30.81	321.88	−3.39
DN106_c0_g1_i6	CBM48 + GH13_8	DIAMOND	N	242.03	540.92	−1.16
DN1722_c0_g1_i20	PL38	HMMER, dbCAN_sub	Y(1–26)	164.19	670.24	−2.03

^1^ Presence or absence of predicted signal peptide. ^2^ Fragments per kilobase of transcript per million mapped reads.

**Table 3 jof-11-00551-t003:** List of enriched GO terms among significantly downregulated DEGs in *F. velutipes* cultivated under 10,000 ppm CO_2_, with commonly identified genes shown in bold.

Enrichment Tool	GO Term	GO ID	Description	Gene Count	Strength	−log_10_ (FDR)	Identified Genes
**STRING database**	Biological process	GO:0000079	Regulation of cyclin-dependent protein serine/threonine kinase activity	2	2.61	0.0321	**DN1560_c0_g1_i9,** **DN6954_c0_g1_i2**
	Molecular function	GO:0016538	Cyclin-dependent protein serine/threonine kinase regulator activity	2	2.61	0.0157	
		GO:0019901	Protein kinase binding	2	2.39	0.0213	
	Cellular component	GO:0000307	Cyclin-dependent protein kinase holoenzyme complex	2	2.59	0.0078	
**topGO**	Biological process	GO:0000079	regulation of cyclin-dependent protein serine/threonine kinase activity	15	2.33	0.007	DN1560_c0_g1_i4, **DN1560_c0_g1_i9**, DN6292_c0_g1_i2, DN6292_c0_g1_i3, DN6292_c0_g1_i4, DN690_c0_g2_i1, **DN6954_c0_g1_i2,** DN8407_c0_g1_i1, DN8407_c0_g1_i17, DN857_c0_g1_i1, DN9546_c0_g2_i1, DN9546_c0_g2_i2, DN978_c0_g2_i1, DN978_c0_g2_i2, DN978_c0_g2_i6
	Molecular function	GO:0019901	protein kinase binding	15	2.27	0.0058	DN1560_c0_g1_i4, **DN1560_c0_g1_i9,** DN3195_c0_g1_i1,DN5489_c0_g1_i5,DN5489_c0_g1_i8, DN690_c0_g2_i1, **DN6954_c0_g1_i2,** DN8407_c0_g1_i14, DN8407_c0_g1_i17, DN857_c0_g1_i1, DN9546_c0_g2_i1, DN9546_c0_g2_i2, DN978_c0_g2_i1, DN978_c0_g2_i2, DN978_c0_g2_i6

**Table 4 jof-11-00551-t004:** Predicted cyclin-related genes in *F. velutipes* identified by transcriptome analysis.

Predicted Gene ID	Length (Amino Acid)	*e*-Value	Accession No.	Description
DN1046_c0_g1_i2	384	1.20 × 10^−44^	IPR039361	Cyclin
DN1046_c0_g1_i5	568	2.10 × 10^−90^	IPR039361	Cyclin
DN1254_c0_g1_i4	602	7.30 × 10^−92^	IPR039361	Cyclin
DN1560_c0_g1_i4	486	9.10 × 10^−49^	IPR013922	Cyclin PHO80-like
DN1560_c0_g1_i9	470	4.80 × 10^−51^	IPR013922	Cyclin PHO80-like
DN2615_c0_g1_i1	390	2.90 × 10^−123^	IPR050108	Cyclin-dependent kinase
DN2615_c0_g1_i18	200	1.90 × 10^−71^	IPR050108	Cyclin-dependent kinase
DN2662_c0_g1_i1	636	7.50 × 10^−83^	IPR039361	Cyclin
DN2961_c0_g3_i11	284	4.80 × 10^−64^	IPR043198	Cyclin/Cyclin-like subunit Ssn8
DN3855_c0_g1_i2	282	1.60 × 10^−84^	IPR050108	Cyclin-dependent kinase
DN3855_c0_g1_i4	419	3.93 × 10^−178^	IPR045267	Cyclin-dependent kinase 11/PITSLRE, catalytic domain
DN4255_c0_g1_i14	421	7.90 × 10^−53^	IPR039361	Cyclin
DN4440_c0_g1_i2	337	2.20 × 10^−120^	IPR050108	Cyclin-dependent kinase
DN4820_c0_g1_i1	520	2.20 × 10^−156^	IPR039361	Cyclin
DN6293_c0_g1_i18	196	1.10 × 10^−87^	IPR050108	Cyclin-dependent kinase
DN6307_c0_g1_i10	372	0	IPR037770	Cyclin-dependent kinase 7
DN6605_c0_g3_i2	323	1.10 × 10^−52^	IPR043198	Cyclin/Cyclin-like subunit Ssn8
DN6605_c0_g3_i5	346	9.80 × 10^−54^	IPR043198	Cyclin/Cyclin-like subunit Ssn8
DN6605_c0_g3_i7	245	1.20 × 10^−22^	IPR043198	Cyclin/Cyclin-like subunit Ssn8
DN6954_c0_g1_i2	473	1.50 × 10^−50^	IPR013922	Cyclin PHO80-like
DN8407_c0_g1_i14	297	2.70 × 10^−33^	IPR013922	Cyclin PHO80-like
DN950_c0_g2_i2	564	1.63 × 10^−6^	IPR036915	Cyclin-like superfamily
DN978_c0_g2_i1	600	2.70 × 10^−28^	IPR013922	Cyclin PHO80-like
DN978_c0_g2_i2	486	2.50 × 10^−28^	IPR013922	Cyclin PHO80-like
DN978_c0_g2_i6	482	2.40 × 10^−28^	IPR013922	Cyclin PHO80-like

## Data Availability

Raw sequencing reads were deposited in the NCBI Sequence Read Archive (SRA) under the accession numbers SRR28157287–SRR28157292.

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
