# Peer review of "High Carbon Dioxide Concentration Inhibits Pileus Growth of Flammulina velutipes by Downregulating Cyclin Gene Expression"

_jof, 2025, doi:10.3390/jof11080551_

Round 1

Reviewer 1 Report

This study investigates the effects of two COâ‚‚ concentrations (1000 ppm and 10,000 ppm) on the growth of Flammulina velutipes. The authors report that under high COâ‚‚ conditions, the cap size decreases significantly while the stipe elongates. Transcriptome (RNA-Seq) analysis of the pileus under both conditions revealed significantly reduced expression of two cyclin genes involved in the cell cycle under high COâ‚‚. The authors claim this is the first transcriptomic study demonstrating cyclin gene expression changes under high COâ‚‚ in F. velutipes, which may have important implications for understanding COâ‚‚-regulated physiology and cultivation strategies in this species.

Comments:

Lack of Comparative Discussion in Cyclin Gene Analysis (Results 3.4–3.6):

In the entire section analyzing cyclin genes (Results 3.4–3.6), no relevant literature is cited for comparison or discussion. Have similar findings been reported in other edible fungi, where cyclin gene expression is influenced by COâ‚‚ concentration? Furthermore, is there evidence from other fungi, plants, or animals that links cyclin genes to COâ‚‚ response? The manuscript presents only the authors' own results, without contextualizing them within existing literatures. If this study truly represents the first investigation of COâ‚‚-induced regulation of cyclin genes, this novelty should be clearly emphasized in the abstract.

Irrelevance and Lack of Integration of CAZyme Analysis (Results 3.3): The CAZyme analysis presented in Results 3.3 appears unrelated to the later discussion on cell cycle regulation. Is its inclusion intended solely to expand the results section? Additionally, the authors report that some identified CAZymes are involved in the degradation of hemicellulose and pectin—both of which are large macromolecules. Why would such macromolecule-degrading enzymes be expressed in the pileus part? This point requires clarification and should be better integrated with the overall biological context of the study.

Figures 3 and 4 should be combined into a single composite figure for better visual clarity and coherence.

In Figure 5, the data points appear incomplete.

Misplaced Transcriptome Overview (Results 3.4): The analysis of transcriptome data begins in section 3.2, yet a general description of the transcriptomic data is presented in 3.4. This overview should be moved earlier, preferably to section 3.2, to maintain a logical flow of results.

Line 106: Please provide a more detailed explanation of the "automated gas regulation system" used in the experiment.

Line 141: Clarify how the qPCR samples were collected. Clearly define the positions and  of "fruiting body" "pileus" and "stipe" in the context of the sampling procedure.

Lines 389–394: The current evidence does not support the conclusion that one cyclin gene functions as a global regulator while the other is tissue-specific. Both genes are expressed throughout the fruiting body. Why would a tissue-specific regulator not function in other tissues?

Author Response

Major comments

Q1. Lack of Comparative Discussion in Cyclin Gene Analysis (Results 3.4–3.6):

In the entire section analyzing cyclin genes (Results 3.4–3.6), no relevant literature is cited for comparison or discussion. Have similar findings been reported in other edible fungi, where cyclin gene expression is influenced by COâ‚‚ concentration? Furthermore, is there evidence from other fungi, plants, or animals that links cyclin genes to COâ‚‚ response? The manuscript presents only the authors' own results, without contextualizing them within existing literatures. If this study truly represents the first investigation of COâ‚‚-induced regulation of cyclin genes, this novelty should be clearly emphasized in the abstract.

Response) We thank the reviewer for this valuable and constructive comment regarding the lack of comparative discussion in the cyclin gene analysis section (Results 3.4–3.6).

  • Relevant comparative literature has already been cited. Studies on Flammulina filiformis and Pleurotus ostreatus have shown that elevated COâ‚‚ broadly downregulates cell cycle-related genes. These references are included in the Introduction (p. 2, lines 66–70), Results (p. 3, lines 169–172; p.10, lines 314-319), and Conclusion (p. 14, lines 461-467).
  • Although plant and animal studies have addressed COâ‚‚-responsive cyclins, their morphology and tissue architecture differ fundamentally from fungi. The pileus, central to our study, has no analog in those systems, making direct comparisons biologically limited. We therefore focused on fungi with similar developmental pathways.
  • To support the functional relevance of the identified cyclins, we also included a comparison with Saccharomyces cerevisiae. The two COâ‚‚-downregulated cyclins in velutipes show strong homology to S. cerevisiae’s PCL1 and PCL2, which regulate the G1/S transition via the Pho85 CDK. This homology is discussed in the manuscript (p. 9, lines 292–297).
  • Finally, in response to the reviewer’s suggestion, we have revised the abstract to clearly state the novelty of our work. The following sentence has been added (page 1, lines 34–38).

Q2. Irrelevance and Lack of Integration of CAZyme Analysis (Results 3.3): The CAZyme analysis presented in Results 3.3 appears unrelated to the later discussion on cell cycle regulation. Is its inclusion intended solely to expand the results section? Additionally, the authors report that some identified CAZymes are involved in the degradation of hemicellulose and pectin—both of which are large macromolecules. Why would such macromolecule-degrading enzymes be expressed in the pileus part? This point requires clarification and should be better integrated with the overall biological context of the study.

Response) We appreciate the reviewer’s insightful comment regarding the CAZyme analysis (Results 3.3).

  1. The inclusion of CAZyme profiling was not intended to simply expand the results section. Rather, our aim was to explore how COâ‚‚ concentration, as an environmental variable, affects the expression of enzymes involved in polysaccharide metabolism—beyond the well-studied context of substrate composition. While many studies have investigated CAZyme regulation in response to different substrates, few have focused on how abiotic factors such as COâ‚‚ influence CAZyme expression. In our study, we observed that several CAZymes, including those involved in the degradation of hemicellulose and pectin, showed significant expression changes under elevated COâ‚‚ conditions. This suggests that COâ‚‚ may play a broader regulatory role in fungal metabolism, potentially influencing nutrient mobilization or structural remodeling in the fruiting body. The expression of such enzymes in the pileus, while unexpected at first glance, may reflect physiological adjustments related to altered gas exchange, internal pressure, or cell wall modification during inhibited development under COâ‚‚ stress. Thus, the CAZyme analysis provides complementary insight into how environmental conditions—beyond substrate—modulate the metabolic landscape of mushrooms. We respectfully note that this result is biologically relevant and not an arbitrary addition.
  2. RNA samples for CAZyme expression analysis were obtained from fruiting body tissues that included both pileus and stipe. While we observed significant changes in the expression of CAZyme genes under elevated COâ‚‚ conditions, we did not perform tissue-specific comparisons nor claim that these enzymes were highly expressed specifically in the pileus. The current dataset reflects bulk expression changes across the upper fruiting body. Future studies with tissue-separated transcriptomic profiling will be valuable to clarify spatial expression patterns and functional localization. However, in response to the reviewer’s comment, we have supplemented the Results section by adding a speculative interpretation related to the point raised (p. 8, lines 257–263).

Detailed comments

Q3. Figures 3 and 4 should be combined into a single composite figure for better visual clarity and coherence.

Response) In response to the reviewer’s suggestion, Figures 3 and 4 have been combined into a single composite figure to improve visual clarity and coherence.

Q4. In Figure 5, the data points appear incomplete.

Response) Thank you for your comment. We have revised and redrawn Figure 4 to ensure that all data points are clearly and fully presented.

Q5. Misplaced Transcriptome Overview (Results 3.4): The analysis of transcriptome data begins in section 3.2, yet a general description of the transcriptomic data is presented in 3.4. This overview should be moved earlier, preferably to section 3.2, to maintain a logical flow of results.

Response) Thank you for the helpful suggestion. In response, we have revised and relocated the general transcriptome overview from section 3.4 to section 3.2 to improve the logical flow of the Results section (p. 5, lines 183–187; p. 9 lines 282–285).

Q6. Line 106: Please provide a more detailed explanation of the "automated gas regulation system" used in the experiment.

Response) We have added a more detailed description of the automated gas regulation system used in the experiment, as requested (p. 3, line 106–107).

Q7. Line 141: Clarify how the qPCR samples were collected. Clearly define the positions and  of "fruiting body" "pileus" and "stipe" in the context of the sampling procedure.

Response) Thank you for the comment. We have revised the manuscript to clarify the qPCR sampling procedure and included definitions of “fruiting body,” “pileus,” and “stipe” in the context of the sampling (pp. 3–4, lines 143–149).

Q8. Lines 389–394: The current evidence does not support the conclusion that one cyclin gene functions as a global regulator while the other is tissue-specific. Both genes are expressed throughout the fruiting body. Why would a tissue-specific regulator not function in other tissues?

Response) Thank you for the insightful comment. We agree that definitive conclusions regarding the functional specificity of the cyclin genes require further experimental validation. While both genes are expressed throughout the fruiting body, our current interpretation (suggesting potential differences in their regulatory roles) is based on observed tissue-dependent expression patterns and is presented as a possible explanation within the manuscript. We have clarified this point and emphasized the need for future functional studies to draw more conclusive insights (p. 13–14, lines 411–414).

Reviewer 2 Report

The authors present the impact of carbon dioxide concentration in pileus growth of Flammulina velutipes. Overall, this work presents the influence that high carbon dioxide concentration has on pileus growth. The obtained results could be useful information for researchers who worked in the similar field.

The authors present the impact of carbon dioxide concentration in pileus growth of Flammulina velutipes. Overall, this work presents the influence that high carbon dioxide concentration has on pileus growth. The obtained results could be useful information for researchers who worked in the similar field.

Some suggestions are as follows:

  1. Fig. 1. Usually, high concentrations of carbon dioxide can inhibit the growth of the pileus and also stimulate the elongation of the stipe. Therefore, it is necessary to supplement the information about the mushroom stipe.
  2. The cyclin genes obtained by RNA-seq should be confirmed by disruption and expression in the wild type, to confirm that they can play an important role in pileus growth.

Author Response

Detailed comments

The authors present the impact of carbon dioxide concentration in pileus growth of Flammulina velutipes. Overall, this work presents the influence that high carbon dioxide concentration has on pileus growth. The obtained results could be useful information for researchers who worked in the similar field.

Some suggestions are as follows:

Q1. Fig. 1. Usually, high concentrations of carbon dioxide can inhibit the growth of the pileus and also stimulate the elongation of the stipe. Therefore, it is necessary to supplement the information about the mushroom stipe.

Response) Thank you for the valuable comment. In general, changes in environmental factors such as COâ‚‚ concentration are known to affect not only pileus size but also stipe elongation. However, in our study, while the pileus diameter of F. velutipes showed a statistically significant difference under different COâ‚‚ conditions, the stipe length did not exhibit a significant change between treatment groups. This observation has been clarified in the revised manuscript (p. 4, lines 166–167).

Q2. The cyclin genes obtained by RNA-seq should be confirmed by disruption and expression in the wild type, to confirm that they can play an important role in pileus growth.

Response) Thank you for the excellent comment. We fully agree that functional validation, such as gene disruption and expression analysis in the wild-type strain, is essential to confirm the roles of the identified cyclin genes in pileus development. In fact, we are currently conducting follow-up studies to address this important aspect, and we sincerely appreciate your thoughtful suggestion.

Reviewer 3 Report

The aim of the article is to study the effects of high carbon dioxide concentration on the pileus development and gene expression of the edible mushroom Flammulina velutipes, offering new insights into how atmospheric COâ‚‚ concentration affects fungal morphogenesis at the genetic level.

Results demonstrated that elevated COâ‚‚ concentration significantly inhibits  pileus development in F. velutipes by downregulating two key cyclin genes involved in  cell cycle progression. These findings provide the first transcriptome-level evidence linking specific cyclin gene expression to COâ‚‚-responsive morphological regulation in F. velutipes. The identification of these genes offers promising targets for future genetic improvement aimed at enhancing COâ‚‚ tolerance and pileus quality in commercial mushroom strains. Additionally, the better understanding of genetic and molecular responses to COâ‚‚ stress can lead to optimization  of  mushroom cultivation cues. ​

The certain study is relevant for the scope of journal. It is presented in a clear and well-structured manner. The experimental design is proper, and methods are described in details. Results are clear and promising and are presented appropriately. I have no recommendations for manuscript improvement, I think it is already suitable for publication

Author Response

Major comments

The aim of the article is to study the effects of high carbon dioxide concentration on the pileus development and gene expression of the edible mushroom Flammulina velutipes, offering new insights into how atmospheric COâ‚‚ concentration affects fungal morphogenesis at the genetic level.

Results demonstrated that elevated COâ‚‚ concentration significantly inhibits  pileus development in F. velutipes by downregulating two key cyclin genes involved in  cell cycle progression. These findings provide the first transcriptome-level evidence linking specific cyclin gene expression to COâ‚‚-responsive morphological regulation in F. velutipes. The identification of these genes offers promising targets for future genetic improvement aimed at enhancing COâ‚‚ tolerance and pileus quality in commercial mushroom strains. Additionally, the better understanding of genetic and molecular responses to COâ‚‚ stress can lead to optimization  of  mushroom cultivation cues. ​

Response) We sincerely thank the reviewer for the thoughtful summary and positive assessment of our manuscript. We appreciate your recognition of the study’s aims and contributions, and are grateful for your constructive comments, which have helped us further improve the clarity and scientific value of the work.

Detailed comments

The certain study is relevant for the scope of journal. It is presented in a clear and well-structured manner. The experimental design is proper, and methods are described in details. Results are clear and promising and are presented appropriately. I have no recommendations for manuscript improvement, I think it is already suitable for publication

Response) We sincerely thank the reviewer for the encouraging and positive feedback. We truly appreciate your recognition of the clarity, structure, and scientific value of our work. Your comments are greatly appreciated and strengthen our confidence in the manuscript.

Reviewer 4 Report

Nicely written Introduction and Materials and Methods.

Section 3.1

It is quite obvious that the elevated carbon dioxide resulted in smaller diameters of the fruiting body but it also significantly increased the number of fruiting bodies. This should be discussed.

Figure 5 – reformat the figure; the proportion taken by each element of the figure doesn’t really make the best use of the real estate on the manuscript.

Line 318 – any particular database used?

Figure 6 – the way in which the information in this figure is presented should be reconsidered as this figure is not straightforward to interpret. Try using another type of visualization commonly used with PSI-BLAST for this figure.

The manuscript is well written but there is a flaw, in my opinion, in this study. The authors went about on the assumption that the size of the pileus is the sole indicator of the wellness of the development. In Figure 1, it is clear that the population under elevated CO2 levels shows significantly increased number of fruiting bodies, which, in certain cases, can be regarded as a more desirable commercial trait. The tone should remain neutral as to focusing on discussing the change brought about by higher levels of carbon dioxide. In order to determine whether such changes are desired or not needs to be evaluated.

To be thorough, I’d like to suggest the authors include a metabolomic investigation to see if the taste/texture of the mushroom is altered and how such alterations effect its demand and desirability (ie. Commercial value).

Detailed comments included in the previous comment.

Author Response

Major comments

Q1. It is quite obvious that the elevated carbon dioxide resulted in smaller diameters of the fruiting body but it also significantly increased the number of fruiting bodies. This should be discussed..

Response) Thank you for the valuable comment. In this study, our primary focus was to investigate and interpret the effects of elevated COâ‚‚ on pileus size from a scientific perspective. We agree that the observed increase in the number of fruiting bodies under elevated COâ‚‚ is also an important phenomenon. As such, we consider this a meaningful point for future investigation and plan to address it in subsequent studies.

Q2. Figure 5 – reformat the figure; the proportion taken by each element of the figure doesn’t really make the best use of the real estate on the manuscript.

Response) Thank you for your comment. We have revised and redrawn Figure 4 to ensure that all data points are clearly and fully presented.

Q3. Line 318 – any particular database used?

Response) Thank you for the comment. We have added the name of the database used in the homology search as requested (p. 10, line 327).

Q4. Figure 6 – the way in which the information in this figure is presented should be reconsidered as this figure is not straightforward to interpret. Try using another type of visualization commonly used with PSI-BLAST for this figure.

Response) Thank you for the helpful suggestion. In response, we have included an additional supplementary figure (Figure S2) that presents the PSI-BLAST results using a more conventional and interpretable visualization format.

Q4. The manuscript is well written but there is a flaw, in my opinion, in this study. The authors went about on the assumption that the size of the pileus is the sole indicator of the wellness of the development. In Figure 1, it is clear that the population under elevated CO2 levels shows significantly increased number of fruiting bodies, which, in certain cases, can be regarded as a more desirable commercial trait. The tone should remain neutral as to focusing on discussing the change brought about by higher levels of carbon dioxide. In order to determine whether such changes are desired or not needs to be evaluated.

To be thorough, I’d like to suggest the authors include a metabolomic investigation to see if the taste/texture of the mushroom is altered and how such alterations effect its demand and desirability (ie. Commercial value).

Response) Thank you very much for your thoughtful and constructive feedback. We agree that the number of fruiting bodies and quality-related traits such as taste and texture are also important factors when evaluating the impact of elevated COâ‚‚, especially from a commercial perspective. While the current study focused primarily on the morphological and transcriptional changes related to pileus development, we fully acknowledge the importance of a more holistic evaluation. We plan to incorporate your suggestion into future studies, including metabolomic analyses to assess quality-related traits and their potential influence on market value.

Round 2

Reviewer 4 Report

The authors did not adequately address the comments and hence the manuscript doesn’t have a clear discussion on the implication of the findings. The data collected is valuable but in order to make the manuscript a good piece of science, a clear “take home message” is rather needed.

See above.

Author Response

Author’s Response to the Academic Editor and Reviewer 4

Editorial Comment

“After careful consideration, the Editor concurs with Reviewer 4’s assessment that the prior revision did not adequately address the major concerns raised in the last round. Specifically: The implications of your findings remain insufficiently articulated. The Discussion lacks a clear, compelling 'take-home message' that situates the work within the broader field. Many of your previous responses deferred substantive issues to 'a future study,' which does not satisfy the requirement to clarify the significance of the present data.”

Response:

We sincerely thank the Academic Editor and Reviewer 4 for their thoughtful and critical feedback on our revised manuscript. We fully acknowledge that our previous revision did not sufficiently address several key concerns, particularly the need for a more clearly articulated “take-home message” and a concrete discussion of the implications of our findings. In response, we have undertaken a thorough and substantive revision of the Abstract, Discussion and Conclusions sections, and we now provide this comprehensive response to clarify how we have addressed each issue raised.

We are sincerely grateful to the Editor and reviewers for guiding us toward a significantly improved version of this manuscript. We hope the revised version now meets the expectations of the journal and respectfully submit it for further consideration.

Point-by-Point Response to Reviewer 4

Q1. “It is quite obvious that the elevated carbon dioxide resulted in smaller diameters of the fruiting body but it also significantly increased the number of fruiting bodies. This should be discussed.”

Response:

Thank you for this valuable observation. In response, we have now included a detailed quantitative description of the increased number of fruiting bodies under high COâ‚‚ conditions (10,000 ppm), supported by Supplementary Figure S4. This observation is discussed in the revised Discussion section (page 14, lines 435–442), where we interpret it as a potential developmental trade-off favoring reproductive output over morphological expansion (i.e., pileus size) under environmental stress.

Q2. “The manuscript is well written but there is a flaw... The authors went about on the assumption that the size of the pileus is the sole indicator of the wellness of the development.”

Response:

We appreciate this critical comment. In the revised Discussion, we have avoided suggesting that pileus size alone reflects developmental success. Instead, we adopt a more nuanced perspective on developmental outcomes, recognizing that high COâ‚‚ conditions may redirect growth resources from morphological expansion to increased reproductive quantity. This revision reflects the complexity of fungal adaptation strategies and avoids overly simplistic interpretations.

Q3. “The tone should remain neutral as to focusing on discussing the change brought about by higher levels of carbon dioxide. In order to determine whether such changes are desired or not needs to be evaluated.”

Response:

We fully agree. The tone in both the Discussion and Conclusion has been revised to reflect neutrality. We now describe COâ‚‚ as a modulator of development rather than as strictly inhibitory, and avoid drawing value judgments about whether the observed changes are beneficial or detrimental.

Q4. “I’d like to suggest the authors include a metabolomic investigation to see if the taste/texture of the mushroom is altered...”

Response:

We greatly appreciate this valuable suggestion. While the current study focuses primarily on transcriptomic responses related to morphological changes under elevated COâ‚‚, we agree that quality traits such as taste, texture, and nutritional content are also important considerations. As such, we have now incorporated this point into the revised Discussion (Section 4, p. 15), noting that future studies should include metabolomic profiling to assess the broader impacts of COâ‚‚ on commercially relevant mushroom traits.

Round 3

Reviewer 4 Report

The authors have addressed the issues. I look forward to future studies that explore other aspects of the effects of elevated levels of carbon dioxide.

N/A.